# MaskFusion: Feature Augmentation for Click-Through Rate Prediction via Input-adaptive Mask Fusion

**Chao Liao, Jianchao Tan, Jiyuan Jia, Yi Guo, Chengru Song**
Kuaishou Technology
{liaochao, jianchaotan, jiajiyuan, guoyi03, songchengru}@kuaishou.com

## Abstract

Click-through rate (CTR) prediction plays important role in the advertisement, recommendation, and retrieval applications. Given the feature set, how to fully utilize the information from the feature set is an active topic in deep CTR model designs. There are several existing deep CTR works focusing on feature interactions, feature attentions, and so on. They attempt to capture high-order feature interactions to enhance the generalization ability of deep CTR models. However, these works either suffer from poor high-order feature interaction modeling using DNN or ignore the balance between generalization and memorization during the recommendation. To mitigate these problems, we propose an adaptive feature fusion framework called MaskFusion, to additionally capture the explicit interactions between the input feature and the existing deep part structure of deep CTR models dynamically, besides the common feature interactions proposed in existing works. MaskFusion is an instance-aware feature augmentation method, which makes deep CTR models more personalized by assigning each feature with an instance-adaptive mask and fusing each feature with each hidden state vector in the deep part structure. MaskFusion can also be integrated into any existing deep CTR models flexibly. MaskFusion achieves state-of-the-art (SOTA) performance on all seven benchmarks deep CTR models with three public datasets.

## 1 Introduction

Click-through rate (CTR) prediction plays an important role in the field of personalized service. Factorization Machine (FM) Rendle (2010) based models are common solutions in recommendation systems. These methods transform the raw high-dimensional sparse features into low-dimensional dense real-value vectors by embedding techniques, and then enumerate all the possible feature interactions, thus avoiding sophisticated manual feature engineering.

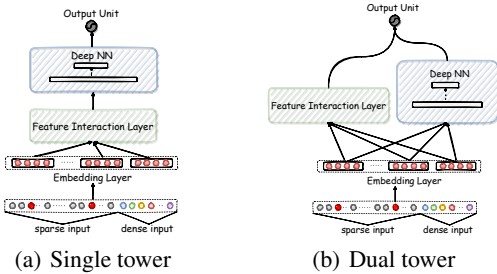

(a) Single tower      (b) Dual tower

Figure 1: Deep learning based Click-Through Rate prediction architecture

Recently, deep CTR models, which incorporate explicit feature interactions module with Deep Neural Networks (DNNs) together, have become a very active research field. As shown in Figure 1, according

---

Corresponding author is Jianchao Tan

to the relative position of DNN and feature interaction layer, the existing deep CTR models can be divided into single tower model and dual tower model Zhang et al. (2021). Single tower models, such as Product-based Neural Networks (PNN) Qu et al. (2016), DLRM Naumov et al. (2019), and so on, can capture the high-order feature interactions to a certain degree, due to their architecture complexity. Dual tower models, such as Deep & Cross Network (DCN) Wang et al. (2017), xDeepfm Lian et al. (2018), and so on, treat DNN as a complementary tool to the feature interactions layer.

Both these two types of deep CTR models utilize the feature interactions layer and the powerful representation ability of DNN to automatically learn high-order feature interactions in explicit and implicit ways, respectively. In fact, these complex structures are designed to address a major challenge in recommendation: **generalization** performance of the CTR models Cheng et al. (2016). For example, the CTR models with strong generalization performance can better explore some feature combinations that have never appeared in historical information, thus recommending new items that users may be interested in.

Currently, some works have attempted to further improve the generalization performance by addressing the problem that DNN is difficult to accurately capture higher-order feature interaction patterns Qu et al. (2018); Rendle et al. (2020). GemNN Fei et al. (2021) introduces a gating mechanism between DNN and the embedding layer of deep CTR models to learn bit-wise feature importance. Features embedding will be fed into DNN after passing through a gating layer instead of being fed into DNN directly. In this way, the DNN can learn more effective feature interaction. MaskNet Wang et al. (2021b) proposed an instance-guided mask, which is generated according to the global information of the instance, to dynamically enhance the informative elements of the hidden states vector by introducing multiplicative operation into DNN. Although these methods optimize the input or hidden states of the deep part in deep CTR models to a certain extent according to the global information of each instance, they did not pay attention to another major challenge in recommendation: the **memorization** ability of the CTR models Cheng et al. (2016). For example, the model can better use the information available in historical data to make relevant recommendations that match user habits, rather than making less relevant recommendations due to over-generalization.

To address the limitation of existing work, we proposed an input-adaptive feature augmentation framework, named MaskFusion, which can incrementally bring non-trivial performance improvements by incorporating various state-of-the-art deep CTR models flexibly and can be trained in an end-to-end manner. Different from the existing methods, by proposing explicit fusion operations, MaskFusion first enhances the memorization ability of the deep CTR models that previous SOTA CTR models have not paid much attention to enhancing. Second, uses the Mask Controller to make a better trade-off between generalization and memorization. Furthermore, incorporated with MaskFusion, CTR models can make predictions for the input instance by using the instance-wise masks to uniquely enhance each feature of this input instance so that the whole model becomes instance-level personalized during both training and inference. We summarize the contributions below:

- We proposed an input-adaptive feature augmentation framework, named MaskFusion, which can capture the interaction between feature embedding and the deep part structure of deep CTR models adaptively and explicitly. MaskFusion framework is general enough to incorporate with other functionalities like Residual Feature Augmentation in DCNv2 Wang et al. (2021a) and Embedding Dimension Search Shen et al. (2020).

- Instance-aware Mask Controller was proposed to dynamically select the feature that needs to be memorized better for prediction task, according to the characteristics and behaviors of different input instances. Thus, can better balance memorization and generalization.

- Comprehensive experiments were conducted on 7 benchmarks over 3 real-world datasets, the convincing results demonstrate the effectiveness and robustness of MaskFusion. Hyperparameters studies demonstrate that MaskFusion is a memory-friendly efficient framework since it achieves better performance with fewer parameters and memories.

## 2    RELATED WORK

DNN begins to benefit recommendation systems because of its powerful feature representation ability. Many works combine explicit feature interaction with DNN in deep CTR models. PNN Qu et al. (2016) introduces a product layer between embedding and DNN to explicitly learn feature interaction.

DeepFm Guo et al. (2017) combines the power of FM (wide part) for recommendation and DNN (deep part) for feature learning. DLRM Naumov et al. (2019) designs a parallelism scheme for the embedding tables to alleviate the limited memory problem. DCN Wang et al. (2017) proposed a Cross Network to learn certain bounded-degree feature interactions explicitly and combined the results of DNN and Cross Network to predict user behaviors. To improve the limited representation ability of DCN in large-scale industrial settings, DCNv2 Wang et al. (2021a) further replaces the cross vector in Cross Network with a cross matrix to make it more practical in large-scale industrial settings. Similarly, xDeepfm Lian et al. (2018) also learns certain bounded-degree feature interactions explicitly through the proposed Compressed Interaction Network (CIN). In addition to learning all possible feature interactions, AutoFIS Liu et al. (2020) automatically searches for important feature interactions in a continuous space to reduce computation costs and noises caused by excessive feature interactions.

The above deep CTR models simply use DNN to automatically model high-order feature interactions, while some other works introduce input-adaptive masks into the models to further augment feature representations. MaskNet Wang et al. (2021b) performs element-wise product both on the feature embedding and hidden states feature vectors in DNN by proposing an instance-guided mask to highlight the important elements. Their intuition is that bringing multiplicative operation into deep CTR models can capture complex feature interaction more efficiently. A similar idea is also proposed in LHUC Swietojanski et al. (2016) in the audio field. Additionally, GemNN Fei et al. (2021) also introduces gating mechanisms to highlight the bit-wise importance of feature embedding before it was fed into DNN. There are some existing works adopted input-adaptive mask-based methods in computer vision Woo et al. (2018); Guo et al. (2022) and natural language processing Dauphin et al. (2017); Kang et al. (2020). VAN Guo et al. (2022) proposes the large kernel attention (LKA) to simultaneously capture spatial and channel-wise long-range correlations adaptively. NMG Kang et al. (2020) adopts a transformer-based policy network to produce task- and domain-adaptive masked context for self-supervised training of language models. In contrast, although we also adopted the mask mechanism, the motivation and the effect are significantly different from these previous works. We apply the mask to augment the features in our proposed fusion layer for two reasons: First, better balance the memorization ability brought by the Fusion Layer and the strong generalization ability from the deep CTR model. Second, make additional explicit interactions between the masked features and the DNN layers to improve the performance. In addition, MaskFusion can be simply integrated into any state-of-the-art deep CTR models to make improvements, as demonstrated in our extensive experiments, while previous methods did not verify whether their designs for DNN and embedding layer are generally applicable to all existing deep CTR models.

## 3 METHOD

In this section, we will introduce the proposed framework MaskFusion in detail. We will first briefly introduce the architecture of one common deep CTR model in Section 3.1; then illustrate each component of MaskFusion in Section 3.2.

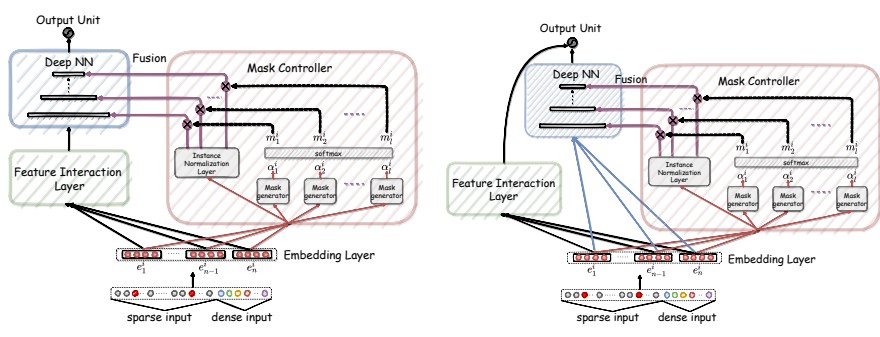

(a) MaskFusion for single tower network    (b) MaskFusion for dual tower network

Figure 2: Illustration of MaskFusion framework for common single and dual tower network. Note that MaskFusion can be easily plugged into any existing network.

### 3.1 DEEP RECOMMENDATION SYSTEM

As illustrated in Figure 1, deep CTR models can be divided into single tower models and dual tower models according to the relative position of *Feature Interaction Layer* and *Deep Neural Network*. *Embedding Layer* embeds the raw feature vectors into a low-dimensional real-value vector. *Feature Interaction Layer* is responsible for capturing feature interactions explicitly (such as inner product, outer product, CIN, CNN Liu et al. (2019), etc.) from the output of *Embedding Layer*. *Deep Neural Network* is usually responsible for extracting higher-order feature interactions implicitly.

### 3.2 MASKFUSION

As illustrated in Figure 2, MaskFusion can be integrated with two types of models. MaskFusion contains three main components, *Mask Controller*, *Instance Normalization Layer*, and *Fusion layer*. To be specific, *Fusion layer* is mainly used to fuse each feature with each layer in the DNN part, so that each layer of DNN can learn feature interaction explicitly. *Mask Controller* generates unique masks for each instance through the *Mask generator* module to guide the *Fusion layer*, thus enabling an adaptive feature fusion mechanism. *Instance Normalization Layer* module normalizes the features that need to be fused with DNN layers. The details will be described in the following sections.

#### 3.2.1 FUSION LAYER

We first introduce Fusion Layer to address the memorization problem which has been described in Section 1. The Fusion Layer can enhance the network's memorization, it explicitly fuses the features into each layer of the DNN through the concatenate operation. In this way, different informative features can be memorized better, since it will allow the features to explicitly participate in the forward inference of the CTR models more directly, rather than unexplainable and complicated transformations of features in the DNN. There are some alternatives to fusion operation, for example, element-wise multiplication in MaskNet Wang et al. (2021b) and GemNN Fei et al. (2021), element-wise addition, and so on. These alternatives cannot explicitly let features be fully exposed to the forward-pass likes ours and have poor explainability for which features contribute more to the prediction performance. Specifically, we fuse the embedding vectors $\mathbf{E}$ with each layer of DNN:

$$\mathbf{h}_t = Relu(\mathbf{W}_t\hat{\mathbf{h}}_{t-1} \ + \ \mathbf{b}_t) \tag{1}$$

$$\hat{\mathbf{h}}_t = Concate([\mathbf{E}, \ Relu(\mathbf{W}_t\hat{\mathbf{h}}_{t-1} \ + \ \mathbf{b}_t)]) \tag{2}$$

where $\mathbf{E} \in \mathbb{R}^{n \cdot d}$, $n$ and $d$ denote the number of feature fields and the dimension of embedding respectively. $\mathbf{W}_t, \mathbf{b}_t$ denote the weight and the bias of $t_{th}$ fully-connected layer in DNN respectively. $\mathbf{h}_t$ is the output of $t_{th}$ fully-connected layer and $\hat{\mathbf{h}}_{t-1}$ is the input of $t_{th}$ fully-connected layer.

#### 3.2.2 MASK CONTROLLER

The aforementioned Fusion Layer module was introduced to enhance memorization. However, the design of only the Fusion Layer also brings two problems at the same time: First, it just utilizes the same network structure and features to characterize all instances identically and ignores the diverse properties among them. The input instances of the recommendation system in the real world always have diverse convergence behaviors, as demonstrated in Zhao et al. (2021). Second, it only considers enhancing the memorization of the model but does not avoid weakening the generalization of the model. To tackle these problems, we propose to adopt an input-adaptive Mask Controller to generate masks based on the global information of each instance and apply the masks to the features in the Fusion Layer. This allows the network to autonomously learn which feature should be utilized better in Fusion Layer according to the diverse properties across different instances during training. After training, a larger mask value means that its corresponding feature in the Fusion Layer should be memorized better and we can know which feature contributes more to the CTR prediction, which makes the prediction with explicit explainability. We use this input-adaptive mask in Fusion Layer to make the network adaptively balance the memorization ability and the generalization ability. To be more specific, given an instance, the embedding vector of all fields feature, represented by $\mathbf{E}$, will be firstly multiplied with this generated unique mask before the fusion processing.

Since MaskFusion is a general framework, the mask generator can be composed of any suitable structure, such as SENet Hu et al. (2018), self-attention Vaswani et al. (2017), MLP, and so on. Here,

we simply use MLP as the mask generator for demonstration purposes and our main contribution lies in designing this general and effective framework.

$$\boldsymbol{\alpha}_t^k = MLP_{\boldsymbol{\phi}_t}(\mathbf{E}) \tag{3}$$

where $\boldsymbol{\alpha}_t^k \in \mathbb{R}^n = [\alpha_{t,1}^k, \cdots, \alpha_{t,n}^k]$ is the mask of $k_{th}$ instance generated by the $t_{th}$ mask generator and $\boldsymbol{\phi}_t$ denotes the parameters of the $t_{th}$ mask generator. As illustrated in Fig 2, the total number of mask generators, denoted as $l$, is equal to the number of layers in DNN.

For the deep part of deep CTR models, embeddings are processed by multiple layers of DNN explicitly, by continuously combining each embedding with other embeddings layer by layer automatically. Thus it can capture potential patterns of the sparse features and will enhance the generalization ability of deep CTR models. After incorporating the Fusion Layer with deep CTR models, feature embeddings are closer to the output layer of DNN and even can directly participate in prediction. As a result, such design not only has a strong generalization ability but also has memorization ability, similar to the wide part design in Cheng et al. (2016). For learning which feature ought to be memorized better automatically and for the purpose of better training convergence, we choose to apply $softmax$ activation functions on all masks to get normalized masks:

$$m_{t,j}^k = \frac{exp(\alpha_{t,j}^k)}{\sum_{t=1}^{l} exp(\alpha_{t,j}^k)}, \; \forall j \in [1,n], \; \forall t \in [1,l] \tag{4}$$

where $\boldsymbol{m}_t^k \in \mathbb{R}^n = [m_{t,1}^k, \cdots, m_{t,n}^k]$ is the mask generated by the $t_{th}$ mask generator from $k_{th}$ instance. Then Eq.(2) will evolve to be:

$$\hat{\mathbf{h}}_t = Concate([\boldsymbol{m}_t^k \mathbf{E}, \; Relu(\mathbf{W}_t \hat{\mathbf{h}}_{t-1} + \mathbf{b}_t)]) \tag{5}$$

The masks are feature-wisely multiplied with $\mathbf{E}$, thus $\mathbf{E} \in \mathbb{R}^{n \cdot d}$ will be reshaped into $\mathbf{E} \in \mathbb{R}^{n \times d}$ and $\mathbf{E} \in \mathbb{R}^{n \times d}$ will be reshaped back into $\mathbf{E} \in \mathbb{R}^{n \cdot d}$ again before the concatenation.

### 3.2.3 INSTANCE NORMALIZATION LAYER

We utilize masks to determine in which layer should the features be fused and how much they will be fused. However, the effect of the mask may become offset by the re-scaling phenomenon. More specially, $m_{t,j}^k \cdot e_j^k$ can produce the same effect as $\frac{m_{t,j}^k}{\varepsilon} \cdot (\varepsilon \cdot e_j^k)$, where $\varepsilon$ denotes a real number, for example, a scalar value of the weight in DNN.

To eliminate this re-scaling phenomenon, a natural method is to use normalization techniques. In MaskFusion, each feature needs to be fused into the DNN first and then be adaptively selected by the mask. The information on each dimension in each feature is very important here, **we should keep bit-wise information while eliminating the re-scaling**. Batch Normalization (BN) and Layer Normalization (LN) do not apply to this scenario, because the calculation of BN counts the information of all instances in a mini-batch and the calculation of LN counts the information of all features in an instance. To address this challenge, we apply Instance Normalization (IN) Ulyanov et al. (2016) on each feature embedding and the IN operation on each feature will be calculated as:

$$IN(\mathbf{e}_j^k) = \boldsymbol{\gamma} \cdot \frac{\mathbf{e}_j^k - \mu_j}{\sqrt{\sigma_j^2 + \epsilon}} + \boldsymbol{\beta} \tag{6}$$

where $\mu_j$ and $\sigma_j$ denote the mean and standard deviation of $j_{th}$ feature of $k_{th}$ instance. Usually, the scale parameter $\boldsymbol{\gamma}$ and shift parameter $\boldsymbol{\beta}$ are set to be trainable to enhance the learning ability of the IN layer. In this paper, we just utilize IN to eliminate the re-scaling phenomenon, so, $\boldsymbol{\gamma}$ and $\boldsymbol{\beta}$ are fixed to 1 and 0 respectively.

**End-to-End Training** MaskFusion is a framework that can be trained in an end-to-end manner. The optimization process is to minimize the objective loss function which is determined according to different recommendation systems tasks, such as click-through-rate prediction (binary classification), user behavior prediction (multi-class classification), and so on. The parameters of MaskFusion and deep CTR models will be updated simultaneously by stochastic gradient descent optimization.

**Finer Granularity version MaskFusion** The dimension of the mask in Eq. (4) is equal to the number of embedding vector outputs from the embedding layer, which means that each dimension in

one feature will share the same mask value. We call it Adaptive Feature-wise MaskFusion (Adaptive FwMF). However, in some complex scenes, a finer mask may be required. Thus we design a finer granularity mask: Adaptive Dimension-wise MaskFusion (Adaptive DwMF). The mask generators in Mask Controller will generate a mask $m \in \mathbb{R}^{n \cdot d}$. In this way, each dimension of each embedding vector will own a unique mask value.

# 4 EXPERIMENTS

In this section, comprehensive experiments are conducted on 7 benchmark models to demonstrate the effectiveness and robustness of the MaskFusion framework over 3 real-world datasets. Due to the page limitation, we refer the readers to the Appendix for more results and analyses.

## 4.1 EXPERIMENT SETUP

**Datasets.** We evaluate the MaskFusion framework on three real-world commercial datasets: *Criteo*, *Terabyte*, and *Avazu*. Due to the page limitation, more details on dataset processing are listed in the Appendix A.

**Evaluation Metrics.** We conduct binary classification (i.e., Click-Through prediction) on the above three datasets and adopt *AUC* and *Logloss* metrics to evaluate all models. Note that an improvement of AUC at 0.001 level will be regarded as a considerable improvement, as also claimed in Zhu et al. (2021); Wang et al. (2021b).

**Baseline Models.** We choose 7 baseline models: IPNN Qu et al. (2016), DeepFm Guo et al. (2017), DCN Wang et al. (2017), xDeepFm Lian et al. (2018), Autoint+ Song et al. (2019), DCN V2 Wang et al. (2021a) and DLRM Naumov et al. (2019). These deep CTR models have various feature interaction layers as described in Section 3.1 and they can be divided into single-tower and dual-tower models. We deploy MaskFusion to these baseline models to show its effectiveness.

## 4.2 PERFORMANCE COMPARISONS

Table 1 summarizes the overall performance of all models. It can be observed that: For all cases in the table, when applying MaskFusion on baseline models, it consistently outperforms the original baseline models over three datasets, no matter whether the model architecture is a single tower or dual tower. It is worth noting that the gap of AUC between different baseline models is very small (even at 0.0001 level), thus **an improvement of AUC at 0.001 level can be considered significant, as claimed in Zhu et al. (2021)**. Compare to the best baseline models DLRM, the relative AUC improvement from DLRM with Adaptive FwMF on Criteo and Avazu is 0.11%, 0.41% respectively. Especially, even on large-scale datasets Terabyte, which consists of 500 million instances, the DLRM model with Adaptive FwMF still outperforms the baseline DLRM model by 0.19% in terms of AUC. Adaptive DwMF realizes feature fusion with a finer-grained adaptive mask manner than Adaptive FwMF. The experimental results show that Adaptive DwMF outperforms Adaptive FwMF for most cases and indeed consistently outperforms overall baseline models.

Table 1: The comparisons of the baseline and two versions of MaskFusion (Feature-wise and Dimension-wise).

| Model | | Criteo | | Terabyte | | Avazu | | Model Type |
| --- | --- | --- | --- | --- | --- | --- | --- | --- |
| | | AUC↑ | Logloss↓ | AUC↑ | Logloss↓ | AUC↑ | Logloss↓ | |
| DCN | Baseline | 0.8046 | 0.4506 | 0.7964 | 0.4197 | 0.8053 | 0.3641 | |
| | Adaptive FwMF | 0.8075 | 0.4480 | 0.7976 | 0.4186 | 0.8144 | 0.3583 | Dual Tower |
| | Adaptive DwMF | **0.8085** | 0.4471 | **0.7985** | 0.4181 | **0.8167** | 0.3573 | |
| Autoint+ | Baseline | 0.8058 | 0.4496 | 0.7968 | 0.4205 | 0.8086 | 0.3621 | |
| | Adaptive FwMF | 0.8073 | 0.4478 | 0.7983 | 0.4183 | 0.8141 | 0.3581 | Single Tower |
| | Adaptive DwMF | **0.8089** | 0.4467 | **0.7989** | 0.4179 | **0.8174** | 0.3562 | |
| DLRM | Baseline | 0.8085 | 0.4469 | 0.7977 | 0.4185 | 0.8145 | 0.3592 | |
| | Adaptive FwMF | 0.8098 | 0.4459 | 0.7992 | 0.4174 | 0.8171 | 0.3570 | Single Tower |
| | Adaptive DwMF | **0.8101** | 0.4456 | **0.7996** | 0.4171 | **0.8186** | 0.3555 | |

To summarize, both types of MaskFusion can surpass the baseline models in terms of AUC and Logloss, which verifies the superiority and the robustness of the proposed framework, in terms of consistently enhancing the deep CTR models with various features interaction manners and various model architectures. **The comparisons of another 4 benchmarks are in Table 6 of the Appendix B.**

## 4.3 ABLATION STUDIES

To give a deeper understanding of the different components of MaskFusion, we perform the ablation studies by gradually adding each component of MaskFusion onto baseline models: **Fusion Layer:** All of the features will be fused to each layer of DNN directly. **Feature-wise Mask:** All of the features will be multiplied by a mask $\beta \in \mathbb{R}^{l \times n}$ before being fused to DNN. This mask is initialized at the beginning of training instead of being generated by our mask controller, thus all instances will share the same mask $\beta$. The mask will be updated during the training process. **Adaptive:** All of the features will be multiplied by a mask before being fused to DNN and the mask will be generated by a mask controller adaptively over the input instance.

The overall performance is shown in Table 2 and we have the following observations:

First, with only *Fusion Layer* added to the baseline, the performances on Criteo and Avazu are even degraded (for DCN, it outperforms the baseline by 0.05% on Criteo in terms of AUC, but it is not significant). On the Terabyte dataset, the performance is similar to the baseline. This phenomenon is reasonable, the Fusion Layer is designed to explicitly enhance the network's memorization ability to remember feature combinations that have appeared in historical information (**memorization is better**). But at the same time, this may also make the network more difficult to make a recommendation if the input feature combinations have never appeared before (**generalization may be worse**).

Secondly, with both *Fusion Layer* and *Feature-wise Mask* added into the baseline, the performance is slightly improved compared to the first case, which with only *Fusion Layer* in the baseline. This demonstrates that the masked features did help solve the generalization problem, however, to a certain degree. We further attribute the reason for small improvements in test AUC to the mask with fixed values. Even though the mask value is updated according to the training data during training, for a mini-batch, all instances in the batch still share the same mask, which is not optimal in CTR prediction scenarios. In industrial scenarios, there are hundreds of millions of users, and each user has different preferences, which also means that the CTR model needs to be more personalized.

Thirdly, with all components incorporated into the baseline, each instance has a unique instance-dependent mask to determine which features should be memorized better for this instance in the Fusion Layer. As expected, this option (called Adaptive FwMF) comprehensively significantly improves all seven baselines on three datasets.

We show the ablations for **another 4 benchmarks in the Appendix C, the observations are same.**

Table 2: Ablation study. Based on the baseline (BL), Fusion (Fu), Feature-wise Mask (FwM), and Adaptive (Ada) is gradually added for ablation studies. ↑ means higher is better and ↓ means lower is better.

| Model | | | | | Criteo | | Terabyte | | Avazu | |
|---|---|---|---|---|---|---|---|---|---|---|
| | BL | Fu | FwM | Ada | AUC↑ | Logloss↓ | AUC↑ | Logloss↓ | AUC↑ | Logloss↓ |
| DCN | ✓ | | | | 0.8046 | 0.4506 | 0.7964 | 0.4197 | 0.8053 | 0.3641 |
| | | ✓ | | | 0.8051 | 0.4503 | 0.7964 | 0.4196 | 0.8046 | 0.3650 |
| | | ✓ | ✓ | | 0.8050 | 0.4503 | 0.7959 | 0.4199 | 0.8062 | 0.3639 |
| | | ✓ | ✓ | ✓ | **0.8075** | 0.4480 | **0.7976** | 0.4186 | **0.8144** | 0.3583 |
| Autoint+ | ✓ | | | | 0.8058 | 0.4496 | 0.7968 | 0.4205 | 0.8086 | 0.3621 |
| | | ✓ | | | 0.8048 | 0.4505 | 0.7967 | 0.4194 | 0.8072 | 0.3631 |
| | | ✓ | ✓ | | 0.8044 | 0.4509 | 0.7969 | 0.4191 | 0.8074 | 0.3630 |
| | | ✓ | ✓ | ✓ | **0.8073** | 0.4478 | **0.7983** | 0.4183 | **0.8141** | 0.3581 |
| DLRM | ✓ | | | | 0.8085 | 0.4469 | 0.7977 | 0.4185 | 0.8145 | 0.3592 |
| | | ✓ | | | 0.8080 | 0.4475 | 0.7980 | 0.4183 | 0.8085 | 0.3624 |
| | | ✓ | ✓ | | 0.8086 | 0.4470 | 0.7983 | 0.4181 | 0.8104 | 0.3611 |
| | | ✓ | ✓ | ✓ | **0.8098** | 0.4459 | **0.7992** | 0.4174 | **0.8171** | 0.3570 |

### 4.4 THE NUMBER OF PARAMETERS AND THE LATENCY

We show the comparisons of the latency and the number of parameters between the baseline models and the baseline models with MaskFusion from two perspectives in Table 3 and 4.

Table 3: The comparisons of the number of parameters and latency time while keeping **the comparable amount** of parameters. †means the number of DNN layers is 3; ‡means the number of DNN layers is 2.

|                        | AUC on Criteo | Params. (M) | latency (ms) |
|------------------------|---------------|-------------|--------------|
| DCN Baseline†          | 0.8042        | 540.72      | 0.15         |
| DCN MaskFusion‡        | 0.8085        | 541.52      | 0.20         |
| Autoint+ Baseline†     | 0.8049        | 541.30      | 0.12         |
| Autoint+ MaskFusion‡   | 0.8087        | 541.41      | 0.15         |
| DLRM Baseline†         | 0.8079        | 540.68.     | 0.06         |
| DLRM MaskFusion‡       | 0.8096        | 541.29      | 0.08         |

As shown in Table 3, MaskFusion only additionally introduces the number of the parameters by 0.1% or 0.2% w.r.t. baseline model, which is negligible. The latency indeed increases due to adding extra parameters and skip-connections in the DNN part of the deep CTR Model, however, it is cost-effective. For example, the latency of MaskFusion increased by 25% while outperforming the baseline model Autoint+ by a significant 0.0038 in terms of AUC. However, compared with the Autoint+ model, the latency of the previous baseline DCN model also increased by 25% while the AUC decreases.

Table 4: The comparisons of the number of parameters and latency time while keeping **the half amount** of parameters. †means the dimension of embedding is 16; ‡means the dimension of embedding is 8.

|                        | AUC on Criteo | Params. (M) | latency (ms) |
|------------------------|---------------|-------------|--------------|
| DCN Baseline†          | 0.8053        | 540.72      | 0.153        |
| DCN+MaskFusion‡        | 0.8070        | 271.57      | 0.125        |
| Autoint+ Baseline†     | 0.8058        | 541.41      | 0.127        |
| Autoint+ +MaskFusion‡  | 0.8067        | 271.03      | 0.131        |
| DLRM Baseline†         | 0.8087        | 540.67.     | 0.064        |
| DLRM+MaskFusion‡       | 0.8081        | 271.69      | 0.078        |

It can be observed from Table 4 that if we reduce the embedding dimension from the original 16 to 8, the parameter size of three shrunk baseline models + MaskFusion will be **50%** of the parameter size of the original three baseline models. Meanwhile, they have comparable or even better AUC and Latency than the three original baseline models. Furthermore, we propose a potential research direction in Appendix I to combine MaskFusion with Embedding Dimension Reduction techniques to further improve the efficiency of MaskFusion, and the preliminary experiments are promising.

In summary, MaskFusion is a cost-effective framework, the observations, analyses, and experiments all show its potential of serving as a plugin to better balance the original baseline model's performance and efficiency, which can be a future exploration.

## 5 TEN INDEPENDENT EXPERIMENTS WITH DIFFERENT SEED

We have kindly considered the robustness of the performance improvements and have reported the mean and standard deviation of AUC for 10 independent runs with different seeds on 3 strong benchmarks in Table 5. The mean improvements range from $1.6e^{-3}$ to $3.9e^{-3}$, and the standard deviations range from $9.8e^{-5}$ to $1.9e^{-4}$. The deviations are significantly smaller than the mean improvements, which demonstrates our improvements are robustly significant. To improve the convincing of the results, we provide the one-tailed independent sample t-test results of the experimental group (Baseline + Adaptive DwMF in table) and the baseline group (Baseline in table) on these three benchmarks. Before the t-test, we first check whether the experimental group and the baseline group have the same variance by Levene's test, if not, we will perform a t-test with the Welch t-test. We can conclude that the p-values corresponding to the t-scores on the three benchmarks are all much

Table 5: The mean and std. of AUC for 10 independent runs with different seeds on the Criteo dataset.

| | Model | AUC | Logloss | Average Precision (AP) | AUC $t$-score | AUC $p$-value |
|---|---|---|---|---|---|---|
| DCN | Baseline | $0.80463 \pm 2.0e^{-4}$ | $0.45067 \pm 1.3e^{-4}$ | $0.6119 \pm 6.3e^{-4}$ | 44.94 | $< 5e^{-4}$ |
| | Adaptive DwMF | $\mathbf{0.80855} \pm 1.9e^{-4}$ | $\mathbf{0.44709} \pm 1.7e^{-4}$ | $\mathbf{0.6168} \pm 4.7e^{-4}$ | | |
| Autoint+ | Baseline | $0.8059 \pm 1.1e^{-4}$ | $0.4496 \pm 2.1e^{-4}$ | $0.6089 \pm 7.0^{-4}$ | 35.33 | $< 5e^{-4}$ |
| | Adaptive DwMF | $\mathbf{0.8089} \pm 9.8e^{-5}$ | $\mathbf{0.4468} \pm 2.5e^{-4}$ | $\mathbf{0.6166} \pm 5.3e^{-4}$ | | |
| DLRM | Baseline | $0.8086 \pm 1.6e^{-4}$ | $0.4469 \pm 1.1e^{-4}$ | $0.6164 \pm 3.6e^{-4}$ | 22.91 | $< 5e^{-4}$ |
| | Adaptive DwMF | $\mathbf{0.8101} \pm 1.4e^{-4}$ | $\mathbf{0.4456} \pm 2.1e^{-4}$ | $\mathbf{0.6202} \pm 2.3e^{-4}$ | | |

lower than the statistical significance threshold of **0.01**. Thus, we can reject the NULL hypothesis and accept the alternative hypothesis that the experimental group outperforms the baseline group with statistical significance for each of the three benchmarks. **We performed t-tests for the other 4 remaining benchmark models in the Table 8 in the Appendix D.**

## 6 ONLINE A/B TESTING

More than just passing industrial A/B testing, MaskFusion has been deployed in multiple industrial scenarios and has achieved significant benefits for online products. It is worth mentioning that in industrial scenarios, the models are often DNN-based models, so it is very suitable to use MaskFusion to further improve performance. We take the A/B testing conducted on the display advertising system as an example. The details of the experiments are as follows:

**Baseline model**    There are a total of 150 features, including 115 user features, 18 item features, 8 combined features, and 9 dense features. The baseline model is a multi-task model, the raw features and the interacted features will be fed into a shared bottom layer which consists of a one-layer MLP with a hidden state size of 1024. The output of the shared bottom layer will be used as the input of 3 towers: IMP, PXR, and CTR, and each tower is an MLP network with the shape of [ 512, 256, 2]. IMP stands for impression rate, PXR stands for the rate of 3 seconds of viewing, and CTR is the click-through rate. It should be noted that only the CTR tower needs to be deployed online, the IMP tower and the PXR tower are used to expand the training dataset and assist in the training of the CTR tower. Therefore, we apply MaskFusion on the CTR tower to improve its performance.

**Performance Improvements**    In the offline testing, baseline+MaskFusion has a 0.003 significant improvement in terms of AUC compared with the baseline model. In the A/B test, the baseline model and the baseline+MaskFusion model will both use 10% traffic respectively, and we mainly focus on the AUC and RPM (Revenue per Mille) metrics. During a month of A/B testing, the baseline+MaskFusion outperforms the baseline with a significant 0.002 AUC improvement and a **10.5%** RPM improvement respectively. It is worth noting that the QPS (Queries per Second) is used to evaluate the efficiency of the online model. In the extreme case testing, the QPS of baseline+MaskFusion only dropped by 6.8% compared to the baseline model. Although this means that we need to use 6.8% more machines to cover all traffic, the cost of the extra 6.8% machines is far less than the 10.5% RPM. Our MaskFusion has been successfully deployed into this online product.

## 7 CONCLUSIONS

In this paper, we proposed an input-adaptive feature augmentation framework, MaskFusion, which can be incorporated with any existing deep CTR models flexibly and enhances their performance stably. In the MaskFusion framework, feature embedding will be multiplied by a mask and then fused to the deep part of deep CTR models through the concatenation operation, so that the DNN can explicitly learn the feature interactions, and the memorization ability of the deep CTR models will also be enhanced effectively. We also proposed a Mask Controller to learn which feature should be memorized better dynamically according to different input instances, thereby making a better trade-off between memorization and generalization automatically. Experiments on 3 real-world datasets and 7 SOTA Deep CTR models demonstrate the effectiveness. Additionally, a latency comparison and a hyper-parameters study show that MaskFusion has the potential to be a memory-friendly and efficient framework since it can achieve better performance with fewer parameters. In the future, we will further explore more efficient and effective dynamic feature augmentation framework designs.

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

# A   IMPLEMENTATION DETAILS

**Datasets.**   For a fair comparison and reproducibility, we adopted the same processing of datasets as in Naumov et al. (2019). *Criteo* consists of almost 45 million instances over 7 days of data and each instance contains 13 numerical features and 26 categorical features. In experiments, the first 6 days' data are used as a training set and the rest as the test set. *Terabyte*, a more challenging dataset, consists of almost 500 million instances over 24 days of data, and each instance contains 13 numerical features and 26 categorical features. In experiments, the first 23 days of data are used as a training set and the last day of data as the test set. *Avazu* consists of almost 40 million instances and each instance contains 23 numerical features. Note that, we remove the *sample_id* field since it is useless for the prediction. The dataset is randomly split by 8:1:1 for training, validating, and testing.

**Training hyper-parameters.**   For a fair comparison, the Deep CTR models integrated with the MaskFusion framework will share the same settings as the baseline models. For numerical features, all of them will be concatenated and transformed into a low dimensional, dense real-value vector by a 4-layers MLP, the number of neurons is [512, 256, 64, 16]. For categorical features, we will embed them into a dense real-value vector with a fixed dimension of 16. For optimization, we utilize an Adagrad optimizer Duchi et al. (2011) with a learning rate of 0.01, and the mini-batch size is 128. The depth of DNN is 3 for all models and the number of neurons is [512, 256, 1]. We simply use the MLP structure as the mask generator, and the structure of each mask generator is [512, 256]. All experiments are conducted on one 2080Ti GPU.

# B   MORE COMPARISONS

We put the comparisons of the baseline model versus the baseline model with two versions of MaskFusion in Table 6. The observations of performance improvements keep the same as in Table 1.

Table 6: The comparisons of the baseline and two versions of MaskFusion (Feature-wise and Dimension-wise).

| Model | | Criteo AUC↑ | Criteo Logloss↓ | Terabyte AUC↑ | Terabyte Logloss↓ | Avazu AUC↑ | Avazu Logloss↓ | Model Type |
|---|---|---|---|---|---|---|---|---|
| IPNN | Baseline | 0.8073 | 0.4483 | 0.7974 | 0.4188 | 0.8118 | 0.3600 | |
| | Adaptive FwMF | **0.8085** | 0.4472 | 0.7984 | 0.4180 | 0.8146 | 0.3581 | Single Tower |
| | Adaptive DwMF | 0.8083 | 0.4475 | **0.7991** | 0.4176 | **0.8179** | 0.3564 | |
| DeepFm | Baseline | 0.8042 | 0.4512 | 0.7953 | 0.4205 | 0.8091 | 0.3623 | |
| | Adaptive FwMF | 0.8054 | 0.4500 | 0.7967 | 0.4194 | 0.8149 | 0.3579 | Single Tower |
| | Adaptive DwMF | **0.8063** | 0.4493 | **0.7976** | 0.4187 | **0.8181** | 0.3562 | |
| xDeepFm | Baseline | 0.8067 | 0.4489 | 0.7972 | 0.4190 | 0.8144 | 0.3584 | |
| | Adaptive FwMF | 0.8080 | 0.4477 | 0.7983 | 0.4181 | 0.8178 | 0.3506 | Dual Tower |
| | Adaptive DwMF | **0.8081** | 0.4475 | **0.7990** | 0.4180 | **0.8189** | 0.3559 | |
| DCN V2 | Baseline | 0.8084 | 0.4475 | 0.7981 | 0.4183 | 0.8082 | 0.3623 | |
| | Adaptive FwMF | **0.8098** | 0.4462 | **0.7991** | 0.4175 | 0.8143 | 0.3579 | Dual Tower |
| | Adaptive DwMF | 0.8091 | 0.4466 | 0.7990 | 0.4175 | **0.8175** | 0.3567 | |

https://www.kaggle.com/c/criteo-display-ad-challenge
https://labs.criteo.com/2013/12/download-terabyte-click-logs/
http://www.kaggle.com/c/avazu-ctr-prediction

### B.1 Analysis of Performance Improvements

As shown in Table 1 and Table 6, we can observe a phenomenon that MaskFusion can achieve more significant orthogonal performance gains on those weaker baseline models compared to those stronger baseline models for all 3 datasets. Since all these models have the same configurations for embedding layers and DNN layers, we attribute this phenomenon to the fact that weaker models tend to have a weaker capacity to model high-order feature interaction in their feature interaction layers and vice versa. As a general feature augmentation framework, MaskFusion proposes an input-adaptive mask-based dynamic interaction mechanism between the embedding layers and the DNN layers that can enhance the feature embedding representation ability and also make up for their poor ability to capture high-order feature interaction. Thus, the weaker the model is, the more performance gains are introduced by MaskFusion.

Both the Criteo dataset and the Terabyte dataset have 39 features (26 sparse features and 13 dense features), while the Avazu dataset has only 23 features and has fewer keys in each embedding table than the former two datasets. It can be observed that the improvement of MaskFusion on Terabyte and Criteo is smaller than that on the Avazu dataset. We attribute this phenomenon to the lack of representation ability of the baseline model in the case of containing a small number of features. After integrating the model with MaskFusion, which can augment these features and better capture the high-order feature interactions, thus enhancing the representation ability of the original model. Thus it leads to more significant orthogonal performance improvements for all 7 baseline models on the Avazu dataset.

Furthermore, both of the above phenomena also corroborate with our statement in Section 4.4 that MaskFusion + baseline model with lower embedding dimensions can achieve comparable performance to the original baseline model.

## C  More Ablation Studies

As illustrated in Table 7, we conduct ablation studies on 4 additional baseline models, including IPNN Qu et al. (2016), DeepFm Guo et al. (2017), xDeepFm Lian et al. (2018) and DCN V2 Wang et al. (2021a). A similar conclusion can be drawn that each component of MaskFusion is indispensable and we can achieve the best performance when combining all of them.

Table 7: More ablation study. Based on baseline (BL), Fusion (Fu), Feature-wise Mask (FwM), and Adaptive (Ad) are gradually added for ablation studies. ↑ means higher is better, ↓ means lower is better.

| | Model | | | | Criteo | | Terabyte | | Avazu | |
|---|---|---|---|---|---|---|---|---|---|---|
| | BL | Fu | FwM | Ad | AUC↑ | Logloss↓ | AUC↑ | Logloss↓ | AUC↑ | Logloss↓ |
| IPNN | ✓ | | | | 0.8073 | 0.4483 | 0.7974 | 0.4188 | 0.8118 | 0.3600 |
| | | ✓ | | | 0.8075 | 0.4480 | 0.7978 | 0.4184 | 0.8086 | 0.3631 |
| | | ✓ | ✓ | | 0.8080 | 0.4476 | 0.7976 | 0.4187 | 0.8116 | 0.3614 |
| | | ✓ | ✓ | ✓ | **0.8085** | 0.4472 | **0.7984** | 0.4180 | **0.8146** | 0.3581 |
| DeepFm | ✓ | | | | 0.8042 | 0.4512 | 0.7953 | 0.4205 | 0.8091 | 0.3623 |
| | | ✓ | | | 0.8042 | 0.4511 | 0.7955 | 0.4204 | 0.8083 | 0.3630 |
| | | ✓ | ✓ | | 0.8044 | 0.4510 | 0.7954 | 0.4206 | 0.8100 | 0.3622 |
| | | ✓ | ✓ | ✓ | **0.8054** | 0.4500 | **0.7967** | 0.4194 | **0.8149** | 0.3579 |
| xDeepFm | ✓ | | | | 0.8067 | 0.4489 | 0.7972 | 0.4190 | 0.8144 | 0.3584 |
| | | ✓ | | | 0.8069 | 0.4487 | 0.7973 | 0.4190 | 0.8134 | 0.3595 |
| | | ✓ | ✓ | | 0.8072 | 0.4484 | 0.7974 | 0.4186 | 0.8129 | 0.3607 |
| | | ✓ | ✓ | ✓ | **0.8080** | 0.4477 | **0.7983** | 0.4181 | **0.8178** | 0.3506 |
| DCNv2 | ✓ | | | | 0.8084 | 0.4475 | 0.7981 | 0.4183 | 0.8082 | 0.3623 |
| | | ✓ | | | 0.8090 | 0.4467 | 0.7986 | 0.4177 | 0.8079 | 0.3630 |
| | | ✓ | ✓ | | 0.8093 | 0.4464 | 0.7986 | 0.4178 | 0.8075 | 0.3632 |
| | | ✓ | ✓ | ✓ | **0.8098** | 0.4462 | **0.7991** | 0.4175 | **0.8143** | 0.3579 |

## D    MORE TEN INDEPENDENT EXPERIMENTS WITH DIFFERENT SEEDS

We conduct additional t-tests experiments on additional 4 models (IPNN, DeepFm, xDeepFm, and DCNv2). We report the mean and standard deviation of AUC for 10 independent runs with different seeds in Table 8. It can be observed that when applying our MaskFusion on these baseline models, it indeed consistently outperforms the original baseline models over three datasets.

Table 8: The mean and std. of AUC for 10 independent runs with different seeds on the Criteo dataset.

| Model | | AUC | Logloss | AP. | AUC $t$-score | AUC $p$-value |
|---|---|---|---|---|---|---|
| IPNN | Baseline | $0.8072 \pm 2.7e^{-4}$ | $0.4485 \pm 3.2e^{-4}$ | $0.6146 \pm 6.6e^{-4}$ | 8.26 | $< 5e^{-4}$ |
| | Adaptive FwMF | $\mathbf{0.8084} \pm 3.1e^{-4}$ | $\mathbf{0.4465} \pm 2.8e^{-4}$ | $\mathbf{0.6163} \pm 5.9e^{-4}$ | | |
| DeepFm | Baseline | $0.8043 \pm 2.0e^{-4}$ | $0.4516 \pm 3.3e^{-4}$ | $0.6088 \pm 4.4e^{-4}$ | 15.62 | $< 5e^{-4}$ |
| | Adaptive DwMF | $\mathbf{0.8062} \pm 2.8e^{-4}$ | $\mathbf{0.4494} \pm 3.2e^{-4}$ | $\mathbf{0.6120} \pm 2.5e^{-4}$ | | |
| xDeepFm | Baseline | $0.8065 \pm 3.7e^{-4}$ | $0.4493 \pm 2.8e^{-4}$ | $0.6130 \pm 6.9e^{-4}$ | 9.86 | $< 5e^{-4}$ |
| | Adaptive DwMF | $\mathbf{0.8083} \pm 3.6e^{-4}$ | $\mathbf{0.4475} \pm 4.0e^{-4}$ | $\mathbf{0.6162} \pm 4.7e^{-4}$ | | |
| DCNv2 | Baseline | $0.8084 \pm 2.1e^{-4}$ | $0.4478 \pm 1.9e^{-4}$ | $0.6157 \pm 5.1e^{-4}$ | 12.99 | $< 5e^{-4}$ |
| | Adaptive FwMF | $\mathbf{0.8099} \pm 2.5e^{-4}$ | $\mathbf{0.4463} \pm 2.4e^{-4}$ | $\mathbf{0.6188} \pm 4.6e^{-4}$ | | |

## E    COMPARE WITH OTHER INPUT-ADAPTIVE BASED MASK METHOD

MaskFusion is a general framework that can be integrated with any deep CTR models and achieve non-trivial improvements. It is unnatural to compare it with specially designed input-adaptive mask-based deep models such as MaskNet [Wang 2021b]. To alleviate the reviewer's concerns, we humbly yet firmly argue that proper experimentation should focus on whether the orthogonal non-trivial performance improvements can be achieved by integrating MaskFusion with the deep CTR model which adopts an input-adaptive mask mechanism, like MaskNet [Wang 2021b]. Therefore, we conducted additional experiments on the recent input-adaptive mask-based baseline model MaskNet. We reimplemented MaskNet, the number of MaskBlock is 5, the reduction ratio is 2, and the architecture type is parallel. For a fair comparison, we trained it using the same hyperparameters and dataset settings as claimed in Appendix A. The results on Criteo and Avazu are as follows:

We report the mean and standard deviation AUC, and Logloss of 10 independent runs with different seeds. If applying our MaskFusion on MaskNet, the performance on Criteo and Avazu datasets can be still further improved by non-trivial 0.18% and 0.25% in terms of AUC, which fully shows that MaskFusion is a general and effective framework that can bring orthogonal non-trivial performance improvement for deep CTR model robustly, even for input-adaptive mask-based models.

Table 9: The comparisons on MaskNet.

| Model | | Criteo | | Avazu | | Model Type |
|---|---|---|---|---|---|---|
| | | AUC↑ | Logloss↓ | AUC↑ | Logloss↓ | |
| MaskNet | Baseline | $0.8081 \pm 1.3e^{-4}$ | $0.4476 \pm 2.2e^{-4}$ | $0.8164 \pm 3.0e^{-4}$ | $0.3582 \pm 5.0e^{-4}$ | Single Tower |
| | Adaptive DwMF | $\mathbf{0.8099} \pm 1.6e^{-4}$ | $\mathbf{0.4453} \pm 2.2e^{-4}$ | $\mathbf{0.8189} \pm 3.4e^{-4}$ | $\mathbf{0.3553} \pm 3.0e^{-4}$ | |

## F    HYPER-PARAMETERS STUDIES

We study how the hyper-parameters impact the performance of DRS models integrated with MaskFusion. We mainly focus on two hyper-parameters: the number of DNN layers and the dimension of the embedding layer. In industrial scenarios, the former mainly affects the inference speed of Deep CTR models, while the latter mainly affects the saving memory of Deep CTR models.

**Number of DNN Layers.** The complexity of the DNN model will increase as the number of layers increases. From Figure 3 to 5, we can observe: For the baseline model, the performance will increase with the depth of DNN and then will degrade because of overfitting. However, the performance of the models incorporated with Adaptive DwMF has smoother improvements as the DNN depth increases.

Meanwhile, the models incorporated with Adaptive DwMF always outperform the baseline as the number of layers increases. This phenomenon demonstrates the effectiveness and robustness of the MaskFusion framework.

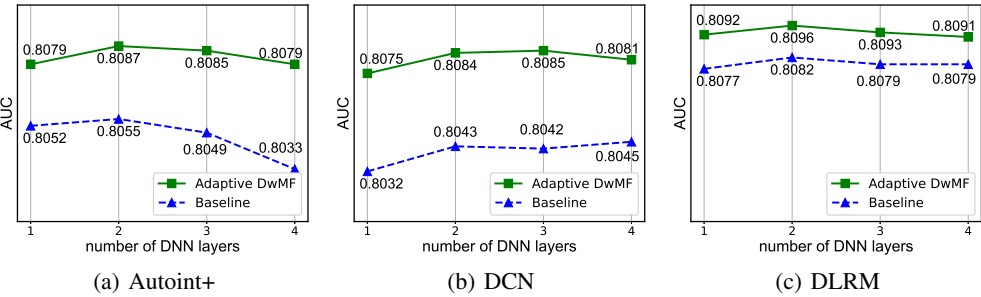

| (a) Autoint+ | (b) DCN | (c) DLRM |

Figure 3: The performance comparison in terms of the number of DNN layers on the Criteo dataset.

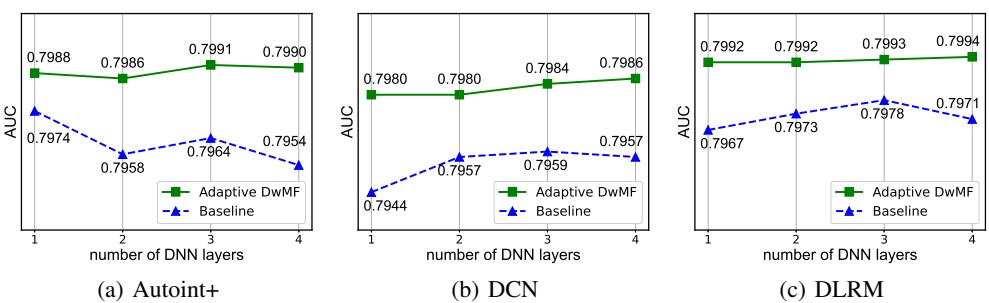

| (a) Autoint+ | (b) DCN | (c) DLRM |

Figure 4: The performance comparison in terms of the number of layers on Terabyte dataset

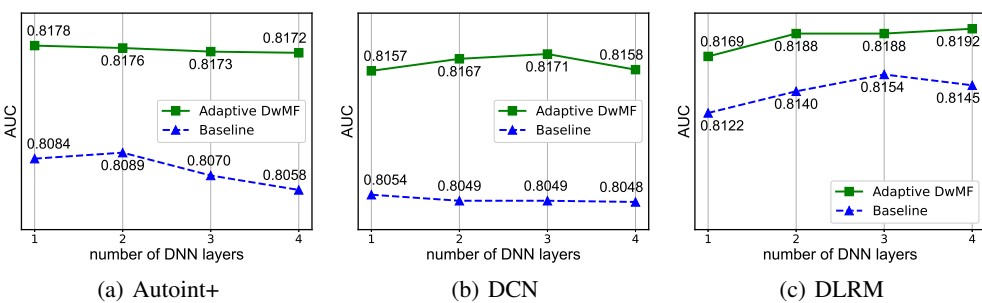

| (a) Autoint+ | (b) DCN | (c) DLRM |

Figure 5: The performance comparison in terms of the number of layers on the Avazu dataset

**Dimension of embedding layer.** Figure 6 to 8 show the impact of the dimension of embedding layer. The performances of all three baseline models with the larger dimension will increase at the beginning. However, except for DLRM, the performance of DCN and Autoint+ will degrade when the dimension becomes larger than 16 and 24 respectively. For models incorporated with Adaptive DwMF, their performances not only consistently outperform the corresponding baseline models but also are continuously improved as the dimension increases from 8 to 32. We attribute this phenomenon to MaskFusion making full use of the low-dimensional dense representation of features: the dynamic interaction mechanism between feature embeddings and DNN structure can enable better embedding table training.

To summarize, MaskFusion shows its effectiveness and robustness over the different hyper-parameters settings. Especially, from the results of Figure 3, we can observe that the models incorporated with

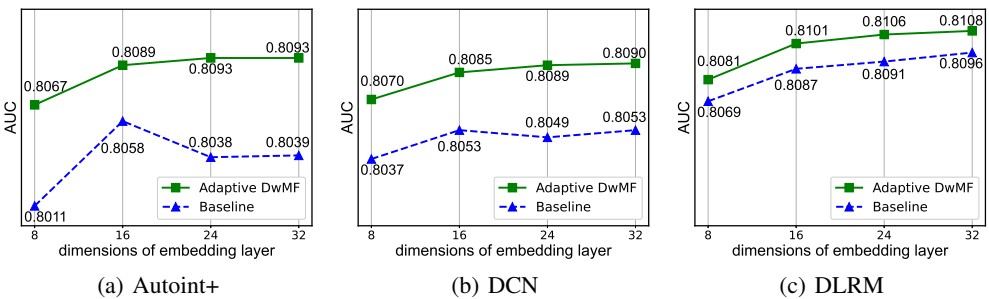

Figure 6: The performance comparison in terms of the embedding layer dimensions on the Criteo dataset.

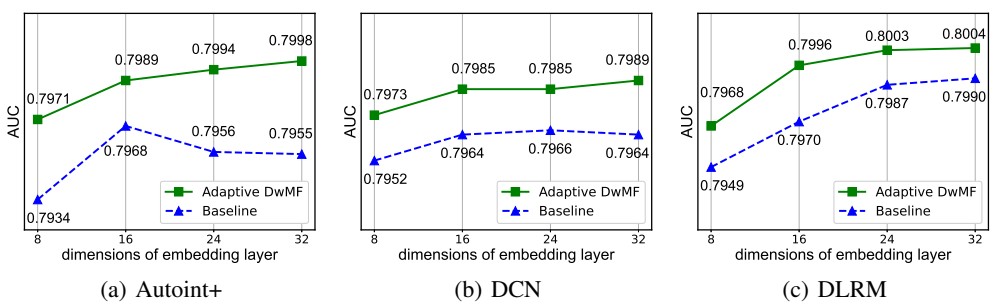

Figure 7: The performance comparison in terms of the embedding layer dimensions on Terabyte dataset

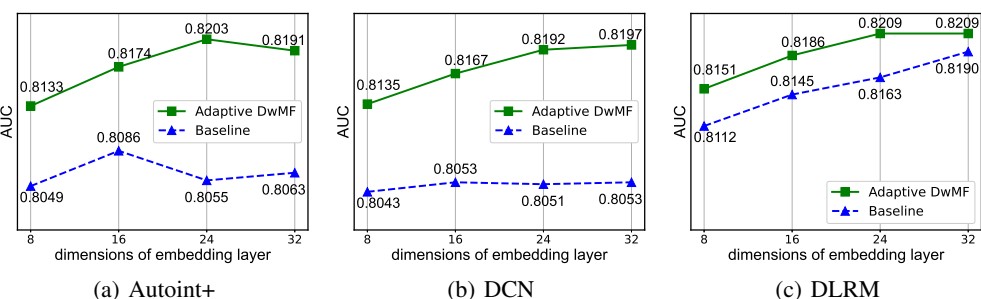

Figure 8: The performance comparison in terms of the embedding layer dimensions on Avazu dataset

MaskFusion under shallower DNN or smaller embedding dimensions scenario can outperform the baseline models that have deeper DNN or larger embedding dimensions setting. **For example, DLRM incorporated with Adaptive DwMF with the embedding dimension value 8 can still outperform the baseline model with the embedding dimension 32 by a non-trivial improvement: 0.06%.** This means that the MaskFusion framework can also be a memory-friendly framework. Numerical features in commercial recommendation systems inevitably lead to huge embedding tables, if the embeddings can be represented only by an 8-dimensional vector, it will save 75% memory consumption compared to a 32-dimensional embedding representation.

From Figure 4 to Figure 8, we can draw similar conclusions. For the baseline model, increasing the model complexity (increasing the number of layers of the DNN or increasing the dimension of the DNN) does not always bring performance improvements to the model; however, there is a smoother improvement in the performance of the model incorporated with MaskFusion as the model complexity increases. This phenomenon is more obvious in *Terabyte*, which is a much larger dataset than *Criteo* and *Avazu*.

As we know, on very large datasets, larger models tend to perform better, however, the baseline models in Figure 4 and Figure 7 do not have a similar phenomenon. We argue the reasons as follows: Simply increasing the network depth may increase the generalization ability of the network. However, the CTR prediction of recommendation scenarios not only requires a strong generalization ability but also needs to be able to make personalized recommendations according to different users' behavior Cheng et al. (2016); Wang et al. (2021b). **This requires the model to take into account both memorization and generalization capabilities, the MaskFusion framework can bring the "input-adaptive" attribute to the model, allowing the model to automatically balance generalization and memorization ability.**

## G    FUSION LAYER FOR MEMORIZATION AND GENERALIZATION

From Section. 4.3, we can conclude: The Fusion Layer is designed to explicitly enhance the network memorization ability to remember feature combinations that have appeared in historical information (memorization is better). But at the same time, this may also make the network more difficult to recommend based on feature combinations that have never appeared before (generalization may be worse). Here, to better support our claim, we provide the training loss and test loss of the baseline model and the baseline model integrated with the Fusion Layer during the training process. As shown in Table. 10, with only the Fusion layer added to the baseline, the training loss is significantly reduced compared to the baseline model (memorization ability on training data is better). However, there are no corresponding reductions for testing loss (generalization ability on test data is not improved, sometimes worse).

Table 10: The training loss and test loss of the baseline model with and without the Fusion Layer.

|  | Criteo training loss | Criteo test loss |
|---|---|---|
| DCN Baseline | 0.4451 | 0.4506 |
| DCN+Fusion Layer | 0.4439 | 0.4503 |
| Autoint Baseline | 0.4447 | 0.4496 |
| Autoint+Fusion Layer | 0.4443 | 0.4505 |
| DLRM Baseline | 0.4434 | 0.4469 |
| DLRM+Fusion Layer | 0.4425 | 0.4475 |

## H    MASKFUSION UNDERSTANDING

The features can interact with the deep part of the deep CTR model dynamically under our MaskFusion framework. Such design will enhance the memorization ability of deep CTR models since the closer the feature is to the output layer, the more directly it can participate in the prediction task Cheng et al. (2016). Here we will show whether the MaskFusion framework captures the feature that needs to be memorized. As shown in Figure 9, Instance A, Instance B, and Instance C are randomly sampled from the dataset, and the mask values for each instance in each layer of DNN are recorded. For instance A, MaskFusion believes that most of the features should be fused to the first and second layers of DNN, instead of the last layer. For instance B, MaskFusion believes that most features should be fused to the last layer instead of the first and second layers. For instance C, most of its features are fused in the first and last layers of the DNN. These different mask values patterns for three different instances can indicate that our MaskFusion framework has successfully learned to choose different best positions where the same features of different instances should be fused, according to the characteristics of different instances, which also indeed dynamically make a trade-off between the generalization and memorization for different instances.

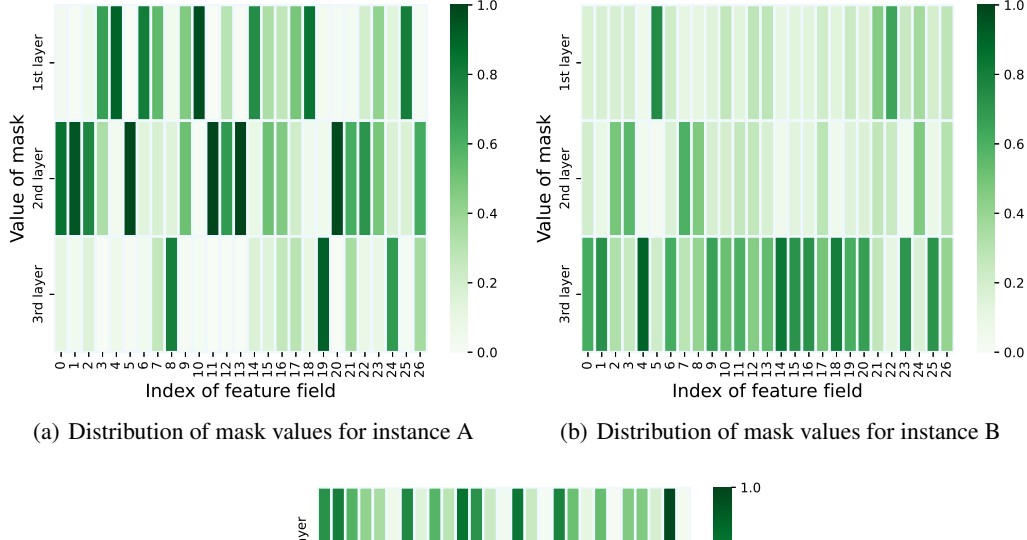

(a) Distribution of mask values for instance A    (b) Distribution of mask values for instance B

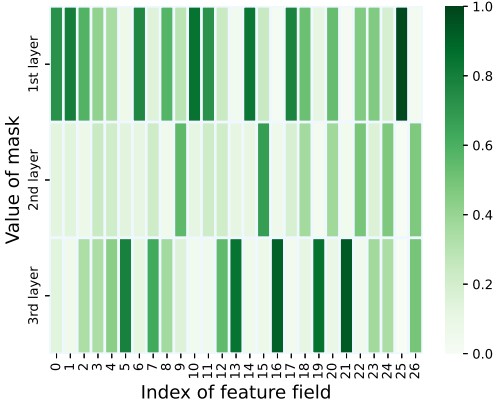

(c) Distribution of mask values for instance C

Figure 9: The mask value distribution of three instances learned on the Terabyte dataset by DLRM integrated with Mask Fusion.

## I    FUNCTIONALITIES EXTENSION

### I.1    MASKFUSION MEETS EMBEDDING DIMENSION REDUCTION.

As a general and easy-to-integrate framework, MaskFusion can be combined with many other methods, such as feature selection Ma et al. (2021); Shen et al. (2020), feature dimension search Shen et al. (2020), etc. Without losing generalization, we use embedding dimensions reduction to prune the dimensions of the fused features and reduce the inference latency time. Deep-learning-based CTR prediction can be formulated by the following function:

$$y = f(\mathbf{W}; \mathbf{E}) \tag{7}$$

where $\mathbf{W}$ denotes the parameters of the Deep CTR model and $\mathbf{E} \in \mathbb{R}^{n \cdot d}$ denotes the embedding vectors ($\mathbf{E}$ also denotes the feature embedding which will be fused to the deep part of the Deep CTR model), $n$ is feature number and $d$ is feature dimension. As shown in Figure 10, to reduce the size of the Deep CTR model, we can prune the dimension of fused feature $\mathbf{E}$ by inserting a gate $\mathbf{g}$:

$$y = f(\mathbf{W}; \mathbf{g} \cdot \mathbf{E}) \tag{8}$$

where $\mathbf{g} \in \mathbb{R}^{n \cdot d}$.

Since we want to reduce the dimension without affecting the model performance, we only remove the corresponding dimension from the feature when $\mathbf{g}$ is exactly 0. Therefore, it is an optimization

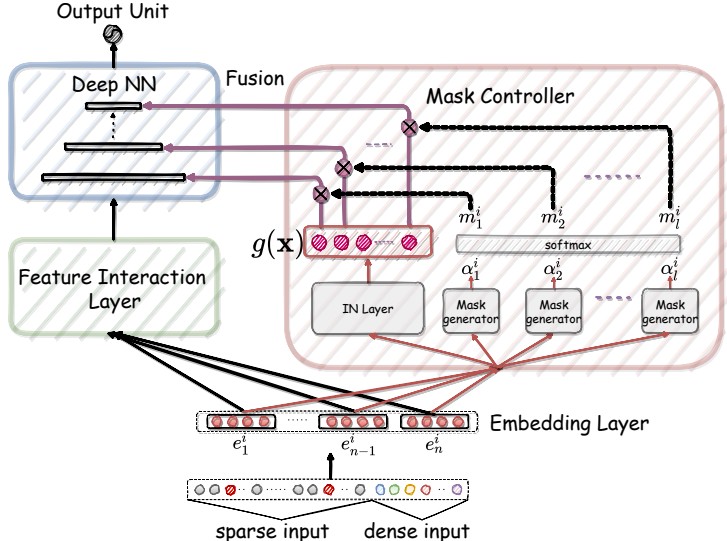

Figure 10: Embedding dimension reduction by inserting smoothed-L0 gate in the MaskFusion framework.

Table 11: The comparisons between the baseline models and SmoothL0 FwMF models. (sparsity: reduce-ratio of parameter number of fused features)

| Model | | Criteo | | | Avazu | | |
|---|---|---|---|---|---|---|---|
| | | AUC↑ | Logloss↓ | sparsity | AUC↑ | Logloss↓ | sparsity |
| IPNN | Baseline | 0.8073 | 0.4483 | -/- | 0.8118 | 0.3600 | -/- |
| | SmoothL0 FwMF | **0.8084** | 0.4474 | 39% | **0.8143** | 0.3585 | 16% |
| DeepFm | Baseline | 0.8042 | 0.4512 | -/- | 0.8091 | 0.3623 | -/- |
| | SmoothL0 FwMF | **0.8055** | 0.4499 | 14% | **0.8132** | 0.3587 | 62% |
| xDeepFm | Baseline | 0.8067 | 0.4489 | -/- | 0.8144 | 0.3584 | -/- |
| | SmoothL0 FwMF | **0.8079** | 0.4479 | 16% | **0.8171** | 0.3523 | 30% |
| DCN | Baseline | 0.8046 | 0.4506 | -/- | 0.8053 | 0.3641 | -/- |
| | SmoothL0 FwMF | **0.8073** | 0.4482 | 58% | **0.8144** | 0.3584 | 56% |
| Autoint+ | Baseline | 0.8058 | 0.4496 | -/- | 0.8086 | 0.3621 | -/- |
| | SmoothL0 FwMF | **0.8072** | 0.4484 | 39% | **0.8125** | 0.3594 | 54% |
| DLRM | Baseline | 0.8085 | 0.4469 | -/- | 0.8145 | 0.3592 | -/- |
| | SmoothL0 FwMF | **0.8097** | 0.4461 | 51% | **0.8170** | 0.3575 | 27% |

problem with the $\ell_0$ constraint. GDP Guo et al. (2021) introduces a smoothed-L0 function to prune the channel of the CNN network. Similarly, we introduce this smoothed-L0 gate function to prune the dimension of the embedding:

$$y = f(\mathbf{W}; g(\mathbf{x}) \cdot \mathbf{E}) \tag{9}$$

$$g(\mathbf{x}) = \frac{\mathbf{x}^2}{\mathbf{x}^2 + \epsilon} \tag{10}$$

where $\epsilon$ is a small positive number, $\mathbf{x} \in \mathbb{R}^{n \cdot d}$. During the training process, the corresponding dimension where $g(\mathbf{x})$ becomes exactly 0 will be pruned from this feature. We present experimental results of applying embedding dimension reduction within the MaskFusion framework in Table 11. In experiments, the parameter $\mathbf{x}$ in the gate function will be updated by SGD Robbins & Monro (1951) optimizer with the momentum parameter 0.9 and the learning rate 0.006. The $\epsilon$ is initialized to 0.1 and will decay by a factor of 0.999, the minimum value is 0.0005. From Table 11, we can conclude: after inserting feature dimension reduction in MaskFusion, Smooth-L0 FwMF consistently outperforms all baseline models with fewer feature dimensions.

Table 12: The comparisons between the baseline models and DCNv2-type FwMF models.

| Model | | Criteo | | Avazu | |
|---|---|---|---|---|---|
| | | AUC↑ | Logloss↓ | AUC↑ | Logloss↓ |
| DCN | Baseline | 0.8046 | 0.4506 | 0.8053 | 0.3641 |
| | DCNv2-type DwMF | **0.8090** | 0.4468 | **0.8134** | 0.3610 |
| Autoint+ | Baseline | 0.8058 | 0.4496 | 0.8086 | 0.3621 |
| | DCNv2-type DwMF | **0.8089** | 0.4467 | **0.8124** | 0.3592 |
| DLRM | Baseline | 0.8085 | 0.4469 | 0.8145 | 0.3592 |
| | DCNv2-type DwMF | **0.8096** | 0.4461 | **0.8171** | 0.3563 |
| IPNN | Baseline | 0.8073 | 0.4483 | 0.8118 | 0.3600 |
| | DCNv2-type DwMF | **0.8086** | 0.4474 | **0.8157** | 0.3578 |
| DeepFm | Baseline | 0.8042 | 0.4512 | 0.8091 | 0.3623 |
| | DCNv2-type DwMF | **0.8061** | 0.4497 | **0.8165** | 0.3586 |
| xDeepFm | Baseline | 0.8067 | 0.4489 | 0.8144 | 0.3584 |
| | DCNv2-type DwMF | **0.8079** | 0.4479 | **0.8191** | 0.3551 |

## I.2 MASKFUSION MEETS DCNV2-TYPE MASK CONTROLLER.

MaskFusion is not limited to a specific augmentation way, besides the aforementioned MLP, it welcomes any other methods to generate masks to augment features. Here, we use DCNv2 as the Mask Controller for illustration. The core of DCNv2 can be expressed by $\mathbf{E}_{l+1} = \mathbf{E}_0 \odot (\mathbf{W}\mathbf{E}_l + \mathbf{b}) + \mathbf{E}_l$, where $\mathbf{E}_{l+1}$ denotes the augmented feature, $\mathbf{W}\mathbf{E}_l + \mathbf{b}$ denotes the mask. So, it can be regarded as a feature augmentation method with a residual signal. From Table 12 we can observe: If using a different augmentation method (DCNv2) as the Mask Controller, MaskFusion can still achieve better performance than baselines.

