# OpenReview forum: "MaskFusion: Feature Augmentation for Click-Through Rate Prediction via Input-adaptive Mask Fusion"
_ICLR.cc/2023/Conference — ICLR 2023 poster_

### Official Review · Reviewer_bHvx · 2022-10-24

**Confidence:** 3
**Correctness:** 4
**Technical Novelty And Significance:** 3
**Empirical Novelty And Significance:** 3
**Recommendation:** 5

**Clarity, Quality, Novelty And Reproducibility:**

Clarity is good and the paper is easy to follow.

Quality is high given detailed component wise introduction/analysis, comprehensive experiment results and studies.

Novelty is fair given a lot of incremental changes over Zhao et al 2021.

Reproducibility is fair and there is no notion of code publicity as I can see.

**Strength And Weaknesses:**

Strengths

The paper is well written and quite readable. Experiment is comprehensive with solid ablation study.

Weaknesses

A strong and recent baseline Wang 2021b is not benchmarked.

There is no data points in complicated prod setting such as industrial AB testing. The method is more or less complexity scaling up and prod setting is much more sensitive to the complexity-gain ROI tradeoff.

**Summary Of The Paper:**

This paper proposes to learn instance-aware per-layer masks for embedding fusion in DNN layers. Both Single tower and dual tower architecture are compatible with such DNN layer mask fusion. A careful instance normalization trick is introduced to resolve re-scaling issue. Experiments have demonstrated orthogonal performance improvement over various deep CTR models.

**Summary Of The Review:**

The paper introduces a per-instance musk fusion method to boost personalization capacity of DNN layers in CTR prediction. The experiment so far reads solid, although the gain is not surprising given boosted complexity.

One big concern is its technical contribution beyond empirical gain. Since this is an application domain narrative, this paper could be a good addition if evidence of adoption in industrial environment is given. It's a marginal paper at its current form.

---

### Official Review · Reviewer_aUmA · 2022-10-25

**Confidence:** 3
**Clarity, Quality, Novelty And Reproducibility:** The work is novel, and the quality an…
**Correctness:** 4
**Technical Novelty And Significance:** 4
**Empirical Novelty And Significance:** 3
**Recommendation:** 8

**Strength And Weaknesses:**

S1. The paper is very well-organized and the writing is good. The figures are clear and helpful for understanding the proposed model structure.

S2. The proposed MaskFusion method is simple, intuitive, and effective. It is also general as can be seamlessly applied to existing deep CTR models, and thus may have a broader impact.

S3. The empirical result is promising as the overall improvement is significant among different backbones and datasets.

---

W1. Since the main comparison results are based on sensitive metrics like AUC, it is suggested to provide mean, variance, and statistical significance values to improve the convincing of the results.

W2. It is suggested to provide an analysis of why the benefit of MaskFusion is more significant on some backbones than others, and on some datasets than others in Table 1.

W3. The discussion on the change of inference time in Appendix E.2 is fundamental and is suggested to be put in the main body.

**Summary Of The Paper:**

This paper proposes a method MaskFusion that can be incorporated into existing deep CTR  models for enhancing prediction performance.

This idea is to multiply input feature embeddings with mask vectors, and then concatenate them with each DNN layer in deep CTR models. The mask vectors are dependent on the input feature embeddings, and are normalized among different layers so that different features will be assigned to different DNN layers.

They conduct experiments on three benchmark datasets. The empirical results demonstrate that MaskFusion can be incorporated into different deep CTR  models to improve their performance.

**Summary Of The Review:**

The proposed MaskFusion method is novel, general and effective for CTR prediction problem. Thus I would vote for accept.

---

AFTER AUTHOR RESPONSE:

I've read the authors' responses and the comments from other reviewers.  Basically, I like the paper for its clarity and soundness, and the response has properly addressed my concerns. Thus I would like to keep my score. I'm still glad to discuss with other reviewers if there are any different opinions after the rebuttal.

---

### Official Review · Reviewer_73YJ · 2022-10-29

**Confidence:** 3
**Correctness:** 3
**Technical Novelty And Significance:** 2
**Empirical Novelty And Significance:** 2
**Recommendation:** 3

**Clarity, Quality, Novelty And Reproducibility:**

(Clarity) The paper is well written and clearly presented.

The code should be made public to validate *reproducibility* owing to the fact that implementation details are missing in the text.

*Quality* and *novelty* is not sufficiently convincing (see Strength and Weaknesses).

**Strength And Weaknesses:**

**Strength**

The paper is well written and clearly presented.

**Weaknesses**

It is not clear whether the proposed framework MaskFusion introduce more computational overhead; it lacks necessary discussion and experiments about efficiency.

Although several existing CTR prediction models with input-adaptive masks are discussed in the Related Work section, experiments comparing these methods are also necessary. It is not satisfying to summarily claim these previous methods are not verified to be applicable to all existing deep CTR models.

The discussion on related work should not be limited to the CTR prediction models. Input-adaptive masks are also wildly used in other fields such as CV and NLP.

**Typos**

Line 22 of Page 3: “different” → “differ”

**Summary Of The Paper:**

This paper proposes a general framework being able to enhance  a wide variety of existing click-through rate (CTR) prediction models. In order to balance memorization and generalization, an instance-wise gating network is utilized to dynamically select the feature embedding which is fused with the deep representation of each layer. Experiments across several benchmarks (existing CTR prediction models) and several CTR prediction datasets demonstrate the effectiveness and generality of the proposed framework.

**Summary Of The Review:**

See “Strength And Weaknesses” and “Clarity, Quality, Novelty And Reproducibility”.

---

### Official Review · Reviewer_Ug5p · 2022-11-01

**Confidence:** 3
**Clarity, Quality, Novelty And Reproducibility:** The paper is quite straightforward to…
**Correctness:** 3
**Technical Novelty And Significance:** 2
**Empirical Novelty And Significance:** 2
**Recommendation:** 5

**Strength And Weaknesses:**

Pros:
- Approach is general and can be applied to several architectures.
- Experimental results are promising.

Cons:
- Concatenating multiple embeddings at each layer of the MLP might explode the number of parameters. The authors mentioned memory saving for certain cases, but I would suggest with existing models by keeping the number of total parameters comparable for a more complete comparison.
- While consistent, the experimental lift in ROCAUC are tiny, I would recommend including average precision as well to better gauge these comparisons.


**Summary Of The Paper:**

The authors focus on the problem of improving the balance between memorization and generalization for CTR prediction, by explicitly incorporating the initial embedding layers into the final layers of typical single and dual tower models.

In particular, each embedding from the embedding layer has a softmax mask determining its contribution to each subsequent layer, and such embedding is concatenated in each layer reweighted by the corresponding mask.

The approach is reminiscent of skip-connections in that it facilitates information flow across many layers. However the embeddings are actually concatenated to the existing layers.

The experimental results are conducted on three realworld dataset and show lift across the board.

**Summary Of The Review:**

While simple, the approach presented show promise in the included experimental results. However, while consistent, the improvements are small and the approach can substantially increase the total number of parameters of the model.

---

### Decision · Program_Chairs · 2023-01-20

**Decision:**

Accept: poster

**Justification For Why Not Higher Score:**

The improvements are small although consistently. In addition, the proposed approach can substantially increase the total number of model parameters.

**Justification For Why Not Lower Score:**

Authors have addressed the concerns from reviewers

**Metareview: Summary, Strengths And Weaknesses:**

This paper aims to improve the balance between memorization and generalization for CTR prediction. The proposed method is generic, which integrates the initial embedding layers into the final layers of typical single and dual tower models. It evaluates the experimental results by using three real-world datasets and demonstrates it performs better than existing methods. It is a well written paper and easy to follow.

**Note From Pc:**

if the above contains the word "oral" or "spotlight" please see: "oral" presentation means -> notable-top-5% and "spotlight" means -> notable-top-25%. As stated in our emails, we are disassociating presentation type from AC recommendations

**Summary Of Ac-Reviewer Meeting:**

Reviewers discussed whether their concerns have been addressed by reviewers sufficiently.

After receiving comments from all reviewers, authors have provided detailed responses and extensive benchmark experiments including industrial A/B tests immediately and proactively.

Reviewers have read thru the authors' rebuttal which have cleared their concerns and thus reviewers support the acceptance